# Exploring Predictors of Self-Perceived Cardiorespiratory Fitness ≥ 5 Years Beyond Breast Cancer Diagnosis: A Cross-Sectional Study

**DOI:** 10.3390/healthcare13070718

**Published:** 2025-03-24

**Authors:** Francisco Álvarez-Salvago, Maria Figueroa-Mayordomo, Cristina Molina-García, Sandra Atienzar-Aroca, Clara Pujol-Fuentes, José Daniel Jiménez-García, Palmira Gutiérrez-García, Rosario Ching-López, Jose Medina-Luque

**Affiliations:** 1FIBIO Research Group, Department of Physiotherapy, Faculty of Health Sciences, European University of Valencia, 46010 Valencia, Spain; francisco.alvarez2@universidadeuropea.es (F.Á.-S.); sandra.atienza@universidadeuropea.es (S.A.-A.); clara.pujol@universidadeuropea.es (C.P.-F.); gutierrezgarciapalmira@gmail.com (P.G.-G.); jose.med.luque@gmail.com (J.M.-L.); 2Department of Health Sciences, Faculty of Health Sciences, University of Jaén, 23071 Jaén, Spain; jdjg@uma.es; 3Faculty of Physiotherapy, Podiatry and Occupational Therapy, Catholic University San Antonio-UCAM, 30107 Murcia, Spain; cmolina799@ucam.edu; 4Department of Dentistry, Faculty of Health Sciences, European University of Valencia, 46010 Valencia, Spain; 5Department of Radiation Oncology, Virgen de las Nieves University Hospital, 18014 Granada, Spain; r.ching.lopez@gmail.com; 6Translational Brain Research, German Center for Neurodegenerative Diseases DZNE, 81377 Munich, Germany

**Keywords:** exercise oncology, rehabilitation, perceived physical fitness, health-related quality of life, long-term survivorship

## Abstract

**Background/Objectives**: This study aimed to examine the association between self-perceived cardiorespiratory fitness and health outcomes in long-term breast cancer survivors (LTBCSs) and identify possible predictors in women at least 5 years post-diagnosis. **Methods**: A cross-sectional study was carried out involving 80 LTBCSs, divided into three groups according to their self-reported cardiorespiratory fitness levels: very poor/poor (1–2), average (3), and good/very good (4–5). Sociodemographic and clinical data were collected, and this study analyzed variables measured at least five years after diagnosis, focusing on various factors including physical fitness, physical activity (PA) levels, cancer-related fatigue (CRF), mood, pain, and health-related quality of life (HRQoL). ANOVA, Mann–Whitney U, and chi-square tests were performed, along with correlation and multiple regression analyses. Cohen’s *d* was used to calculate effect sizes. **Results**: Among the 80 LTBCSs, 35% reported very poor/poor self-perceived cardiorespiratory fitness, 35% reported average levels, and 30% reported good/very good levels. Individuals with lower self-perceived cardiorespiratory fitness levels showed significant declines in physical fitness, greater physical inactivity, increased CRF, higher pain levels, and a poorer HRQoL (*p* < 0.05). Regression analysis identified “self-perceived muscle strength” (β = 0.40; *p* < 0.01) and “nausea and vomiting” (β = −0.37; *p* < 0.01) as significant predictors of higher self-perceived cardiorespiratory fitness (adjusted r^2^ = 0.472). **Conclusions**: These findings highlight the importance of self-perceived cardiorespiratory fitness as a relevant indicator of health outcomes in LTBCSs. Given its association with physical fitness, sedentary behavior, CRF, pain, and HRQoL, assessing patients’ perceptions may provide valuable insights for developing tailored rehabilitation strategies. Future interventions should consider both subjective and objective measures to optimize the long-term health and quality of life in this population.

## 1. Introduction

Breast cancer (BC) statistics indicate that the incidence rate among women was 23.8%, while the mortality risk reached 15.4% worldwide in 2022 [1]. The introduction of mass mammographic screening has facilitated the earlier detection of smaller and less aggressive tumors. Combined with therapeutic advancements, a significant increase has been seen in the number of BC survivors in recent decades [1]. However, as the survivor population grows, making BC a long-term condition for many, the long-term (i.e., beyond five years since BC diagnosis) adverse effects of treatment have gained increasing attention. This has led to a broad body of scientific research aimed at assessing the health status of these long-term breast cancer survivors (LTBCSs) and developing rehabilitation programs to address the sequelae that persist five years post-diagnosis [2]. Despite these efforts, some aspects of long-term survivorship remain underexplored, particularly the impact of self-perceived physical fitness and its various components. Gaining a deeper understanding of these factors is crucial for designing more effective rehabilitation strategies to enhance the long-term well-being of this population.

In this sense, self-perceived physical fitness serves not only as a health indicator but also as a predictor of mortality [3]. This concept encompasses multiple components, such as cardiorespiratory fitness, muscular strength, agility, speed, and flexibility [4], while also reflecting an individual’s physical capabilities and their ability to handle daily challenges with energy and efficiency [4]. As a relatively stable measure, it provides valuable insight into habitual physical activity (PA) levels [5].

Among the various components of physical fitness, cardiorespiratory fitness stands out as a key indicator of overall health and health-related quality of life (HRQoL) [6]. Additionally, studies in various populations, including adolescents, adults, and short-term BC survivors, have linked higher self-perceived cardiorespiratory fitness to improved physical and mental health as well as to an enhanced HRQoL [7,8,9,10]. It is also associated with a greater likelihood of engaging in PA and adopting healthier lifestyles [8]. However, despite these findings, research specifically examining self-perceived cardiorespiratory fitness in LTBCSs and its potential links to their physical, emotional, and mental well-being remains limited.

Cardiorespiratory fitness can be assessed using various methods. Laboratory-based assessments, such as maximal oxygen uptake (VO₂max) tests, are considered the gold standard due to their precision and objectivity. Notwithstanding, these methods are resource-intensive, requiring specialized equipment, trained personnel, and significant time investment, making them impractical for large-scale or routine evaluations [11]. As a practical alternative, self-reported measures have been developed to estimate cardiorespiratory fitness without requiring exhaustive testing. Research has shown that non-exercise models incorporating variables such as age, sex, body mass index, resting heart rate, and both self-reported physical fitness and PA levels can reliably predict cardiorespiratory fitness [12,13,14,15]. These self-assessment tools offer a feasible approach for large-scale screenings, particularly in populations where traditional testing is impractical, thereby informing public health strategies and targeted interventions.

Finally, considering the existing evidence supporting the benefits of cardiorespiratory fitness, along with the utility of self-perceived measures as a viable alternative when direct assessments are not feasible [11,12,13,14,15,16,17], it is notable that the relationship between self-perceived cardiorespiratory fitness and health outcomes in LTBCSs remains largely unexplored. Future studies should aim to fill this gap by examining how self-perceived cardiorespiratory fitness influences various health parameters and whether targeted interventions could enhance both perceived and actual fitness levels in this population. Therefore, the purpose of this study was to explore the association between self-perceived cardiorespiratory fitness and health outcomes in LTBCSs as well as identify possible predictors in women at least 5 years post-breast-cancer diagnosis.

## 2. Materials and Methods

### 2.1. Design and Participants

This cross-sectional study was conducted between late 2022 and early 2023 at the Sport and Health Joint University Institute (iMUDS), following the Declaration of Helsinki (14/2017) [18] and approved by the Biomedical Research Ethics Committee of Granada (CEIm) (1038-N-16 I.P/07/26/2018).

A total of 80 LTBCSs participated in this study. Eligible participants were identified and recruited through the oncology department of the University Hospital Complex of Granada during the specified study period. Additional details can be found in Appendix A.

Participants were informed about the study objectives via telephone and given the opportunity to ask questions before providing written informed consent during an in-person assessment session conducted by a trained physiotherapist from the research team. Each session lasted approximately one hour and took place on a single day, during which participants completed a comprehensive assessment dossier. This dossier included all questions related to demographic and clinical variables, as well as validated questionnaires assessing the study variables.

A trained physiotherapist, experienced in oncology patient assessment, supervised the process, ensuring the participants understood the questionnaires and providing assistance if needed. The physiotherapist overseeing the assessment was different from the team member responsible for digitizing the responses, ensuring data integrity. Additionally, the statistical analysis was conducted by a separate researcher, independent from both the assessment and data entry processes.

### 2.2. Eligibility Criteria and Group Classification

#### 2.2.1. Inclusion Criteria

Female sex.Age ≥ 18 years.Diagnosis of stage I–IIIa breast cancer at least five years prior to enrollment.No participation in structured fitness-related assessments within the past three months (to avoid bias in self-reported fitness levels).

#### 2.2.2. Exclusion Criteria

Any condition preventing the completion of the assessment protocol.Inability to understand the study procedures.

Participants were classified based on self-perceived cardiorespiratory fitness using the International Fitness Scale (IFIS), a validated instrument which is rated on a 1–5-point scale. Based on the prior literature [14,19], participants were grouped as follows: very poor/poor (1–2), average (3), and good/very good (4–5).

The necessary sample size was estimated using G*Power (Version 3.1.9.7) for a comparison of three independent groups. Assuming a medium effect size (f = 0.25), an alpha level of 0.05, and a power of 0.80, the required total sample size was 72 participants (24 per group). Finally, we recruited 80 participants, ensuring adequate statistical power for the analyses.

### 2.3. Variables

#### 2.3.1. Demographic and Clinical Data Collection

Information was obtained through organized interviews using a customized questionnaire designed to collect sociodemographic and clinical data. Clinical variables included the time since diagnosis, tumor stage, family history of BC, surgical procedures, types of treatment received, current medications, presence of metastasis or recurrence, menopausal status, utilization of psychological or physiotherapy services, and lifestyle factors such as tobacco and alcohol use.

#### 2.3.2. Physical Fitness

Self-perceived fitness levels were evaluated using the IFIS with responses measured on a 5-point Likert scale, ranging from 1 (very poor) to 5 (very good). The tool consists of five core questions that assess overall physical fitness along with specific aspects such as cardiorespiratory endurance, muscular strength, speed/agility, and flexibility relative to peers. This questionnaire has demonstrated reliability, with a Cronbach’s alpha of 0.80 [14].

#### 2.3.3. Physical Activity Level

The Minnesota Leisure Time Physical Activity (MLTPA) questionnaire was used to assess the average frequency and total hours dedicated to PA over the past week. This tool has demonstrated high reliability, with an intraclass correlation coefficient (ICC) of 0.95 [20]. The evaluation was based on a preselected list of specific physical activities. Energy expenditure was determined by merging the reported weekly duration (in hours) of each activity by its corresponding metabolic equivalent of task (MET) value [21], which reflects the energy cost of an activity. A higher final score indicates a greater amount of time spent engaging in PA each week.

#### 2.3.4. Cancer-Related Fatigue

The Piper Fatigue Scale (PFS) was used to assess cancer-related fatigue (CRF). This is a 22-item tool, which measures CRF across four main dimensions: behavioral severity, affective, sensory, and cognitive/mood. The overall score from this scale indicates the level of CRF, with higher scores reflecting greater fatigue [22,23]. The reliability of this tool has been confirmed, with a Cronbach’s alpha of 0.86 [24]. Based on previous research, two scoring models (Models A and B) were suggested as effective for classifying CRF severity. In Model A, CRF is categorized into the following: 0 = none, 1–3 = mild, 4–6 = moderate, and 7–10 = severe; Model B categorizes it as 0 = none, 1–2 = mild, 3–5 = moderate, and 6–10 = severe [22,23]. Importantly, moderate CRF, irrespective of the model used, is considered clinically significant [25]. Patients who screen positive for moderate-to-severe CRF should receive a diagnostic evaluation to identify any underlying or comorbid conditions that may require treatment [26].

#### 2.3.5. Mood State

The Scale for Mood Assessment (EVEA) was used to evaluate four different mood dimensions, with a Cronbach’s alpha between 0.88 and 0.93 [27]. This tool consists of 16 items, each rated on a Likert scale from 0 to 10. The scores for each mood dimension—sadness/depression, anxiety, anger/hostility, and happiness—are calculated by averaging the scores of the items within each respective category. Higher final scores indicate a greater intensity of the corresponding mood dimension.

#### 2.3.6. Pain Measures

Pain intensity was evaluated using the Visual Analogue Scale (VAS), a 10 cm scale known for its high reliability (ICC = 0.97) [28], where 0 signifies “no pain” and 10 represents the “worst possible pain.” Participants were asked to rate the pain in both their affected and unaffected arms at the time of the assessment. For those with bilateral BC where both arms were impacted, the arm deemed “affected” was identified based on these factors: (1) the patient’s subjective pain report when comparing the two arms, (2) the level of surgical intervention, and (3) the presence of lymphedema or other post-surgical complications. Additionally, the Brief Pain Inventory (BPI) short form, with Cronbach’s alpha values ranging from 0.87 to 0.89 [29], was employed to measure pain intensity (severity) through four questions and its impact on daily life (interference) through seven questions. Higher scores reflect greater pain intensity and more significant interference with daily activities.

#### 2.3.7. Health-Related Quality of Life

Two validated tools were used to assess HRQoL: the EORTC QLQ-C30 (version 3.0) and its BC-specific module, the QLQ-BR23. The reliability of these tools has been established, with Cronbach’s alpha values ranging from 0.46 to 0.94 [30,31]. Each question is answered on a 4-point Likert scale (1 = not at all, 4 = very much) and then transformed to a scale of 0 to 100. When interpreting the results, higher scores on the functional and global HRQoL scales indicate better health, while higher scores on the symptom scales suggest a greater symptom impact. Additionally, a summary score for the QLQ-C30 was calculated by combining the scores from 13 scales and items, excluding the global health status and financial impact scales. For the summary score, higher values correspond to better HRQoL [32].

#### 2.3.8. Statistical Analysis

Data analysis was performed using IBM SPSS Statistics (version 27.0, Armonk, NY, USA), with significance set at *p* < 0.05 and a 95% confidence interval (CI). Normality was assessed via the Kolmogorov–Smirnov test (*p* > 0.05). For normally distributed continuous variables, ANOVA compared the three self-perceived cardiorespiratory fitness groups: very poor/poor (1–2), average (3), and good/very good (4–5). Non-normally distributed variables were analyzed using the Kruskal–Wallis test, with Mann–Whitney U tests for pairwise comparisons. Chi-square tests were used for categorical and ordinal variables. Effect sizes (Cohen’s *d*) were categorized as negligible (*d* = 0–0.19), small (*d* = 0.2–0.49), moderate (*d* = 0.5–0.79), large (*d* = 0.8–1.19), and very large (*d* ≥ 1.20) [33].

To analyze the relationship between self-perceived cardiorespiratory fitness (measured by IFIS) and other variables, Spearman correlation analysis was used. Stepwise multiple regression identified factors influencing its variability. Variables were included if they correlated significantly with the dependent variable and had inter-variable correlations below 0.70 to avoid collinearity [34,35]. A forward selection method sequentially added significant predictors, assessing statistical significance at each step. Standardized β coefficients were calculated for the final model. Non-normally distributed variables, identified via the Kolmogorov–Smirnov test, were transformed using logarithmic and square root methods to meet regression assumptions.

## 3. Results

### 3.1. Demographic and Clinical Characteristics

Following similar approaches used in previous studies [14,19], participants were classified as having very poor/poor (35%), average (35%), or good/very good (30%) self-perceived cardiorespiratory fitness. No significant differences were observed between the groups regarding the demographic and clinical characteristics of the 80 LTBCSs based on their self-reported cardiorespiratory fitness levels.

The average age of the participants with very poor/poor self-perceived cardiorespiratory fitness was 47.20 ± 8.26 years, while those with an average level had a mean age of 49.46 ± 7.09 years, and participants with good/very good fitness levels had an average age of 52.04 ± 8.41 years. In the very poor/poor fitness group, 42.8% were on sick leave, 46.4% had undergone a quadrantectomy, and 25% had metastasis. In the average fitness group, 49.3% were on sick leave, 39.3% had undergone a quadrantectomy, and 14.3% had metastasis. Among those in the good/very good fitness group, 25% were on sick leave, 62.5% had undergone a quadrantectomy, and 12.5% had metastasis. Further details on the demographic and clinical characteristics can be found in Table 1.

### 3.2. Physical Fitness

The analysis of physical fitness, as assessed with the IFIS, revealed significant differences between the groups. The comparisons of self-perceived cardiorespiratory fitness levels indicated that LTBCSs with good/very good self-reported fitness exhibited higher scores across all domains compared to both the very poor/poor (U = 98.50 to 131.00; *p* < 0.01; *d* > 1.20) and average (U = 147.50 to 177.50; *p* < 0.01; *d* = 0.81 to 1.10) groups. Furthermore, the very poor/poor group demonstrated significantly lower scores for speed/agility compared to the average group (U = 258.50; *p* = 0.01; *d* = 0.69). A visual summary of these findings is presented in Table 2.

### 3.3. Physical Activity Level

The comparison of MLTPA scores between groups revealed a significant difference between LTBCSs with very poor/poor self-perceived cardiorespiratory fitness and those with good/very good levels. Specifically, a greater proportion of LTBCSs in the very poor/poor group were classified as “inactive” (42.8%), while only 20.8% of those in the good/very good group were categorized as “inactive” (*p* = 0.02). No significant differences were observed between the other group comparisons (*p* > 0.05). Further details are provided in Table 2.

### 3.4. Cancer-Related Fatigue

The analysis of the PFS domains showed significant differences when comparing the very poor/poor and average groups to the good/very good group. LTBCSs with very poor/poor self-perceived cardiorespiratory fitness reported significantly higher values in all domains compared to the good/very good group (U = 138.50 to 170.00; *p* < 0.01; *d* = 0.98 to 1.14). Likewise, the average group had significantly higher values across all domains when compared to the good/very good group (U = 126.00 to 153.00; *p* < 0.01; *d* = 0.29 to 1.19). No significant differences were found between the very poor/poor and average groups (*p* > 0.05). Additional details are available in Table 2.

Regarding the two cut scores, A and B, the analysis revealed that LTBCSs with very poor/poor and average self-perceived cardiorespiratory fitness levels reported significantly higher proportions of “severe” CRF for both cut scores. Specifically, in the very poor/poor group, 14.2% (A) and 28.5% (B) had severe CRF, while, in the average group, 17.9% (A) and 21.4% (B) exhibited severe CRF. In contrast, no LTBCSs in the good/very good group reported severe CRF (A = 0%, B = 0%) (*p* = < 0.01 to 0.03). No significant differences were found when comparing the very poor/poor group to the average group for either of the two cut scores (*p* > 0.05). These findings are summarized in Table 2.

### 3.5. Mood State

The analysis of the EVEA outcomes, assessing mood states, showed no significant differences between groups for any of the three comparisons (*p* > 0.05). Detailed values can be found in Table 2.

### 3.6. Pain

The analysis of pain, based on VAS scores, showed no significant differences between the group comparisons. However, when using the BPI, significant differences were observed for “pain intensity” and “pain interference” between the very poor/poor (U = 164.00 and 154.50; all *p* < 0.01; *d* = 0.94 to 1.12) and average (U = 211.00 and 212.50; all *p* = 0.02; *d* = 0.56 to 0.68) groups in comparison to the good/very good group. Specifically, the good/very good group reported significantly lower values than the other two groups. No significant differences were found between the BPI values of the very poor/poor and average groups. These results are presented in Table 2.

### 3.7. Health-Related Quality of Life

The analysis of HRQoL, measured using the QLQ-C30 and -BR23, showed significant differences across the groups. When comparing the very poor/poor group to the good/very good group, significantly lower values were found in the areas of “role functioning”, “emotional functioning”, “cognitive functioning”, “global health status”, “summary score”, “body image”, “sexual functioning”, and “systemic therapy side effects”. Conversely, higher values were observed for “breast symptoms” and all individual items, except for “diarrhea” and “financial difficulties” (U = 128.00 to 279.00; *p* < 0.01 to 0.03; *d* = 0.64 to 1.10). Furthermore, LTBCSs with very poor/poor self-perceived cardiorespiratory fitness compared to the average group also reported significantly lower levels of “emotional functioning” and “sexual enjoyment”, as well as higher levels of “nausea and vomiting” (U = 280.50 to 293.00; *p* = 0.04 to 0.05; *d* = 0.50 to 0.56). A similar pattern was observed when comparing the average group to the good/very good group, with significantly lower values in the average group for “role functioning”, “cognitive functioning”, “global health status”, “summary score”, “body image”, “sexual functioning”, and all symptom scales, except for “upset by hair loss”. Additionally, the average group had higher values for “nausea and vomiting” compared to the good/very good group (U = 144.50 to 277.00; *p* < 0.01 to 0.05; *d* = 0.29 to 1.10). Further details are available in Table 3.

### 3.8. Correlation and Multiple Regressions Analysis

Spearman’s correlation analysis revealed significant positive correlations between self-perceived cardiorespiratory fitness and several variables, including QLQ-C30 items such as “role functioning”, “emotional functioning”, “cognitive functioning”, “social functioning”, and “fatigue”; QLQ-BR23 items like “sexual functioning” and “systemic therapy side effects”; and IFIS components such as “general physical fitness”, “muscular strength”, “speed/agility”, and “flexibility” (ρ = 0.233 to 0.575; *p* < 0.01 to 0.03). In contrast, significant negative correlations were found between self-perceived cardiorespiratory fitness and the following variables: PFS items like “behavioral/severity”, “affective”, “sensory”, “cognitive/mood”, and “total fatigue score”; QLQ-C30 items including “nausea and vomiting”, “dyspnea”, “insomnia”, “appetite loss”, and “constipation”; QLQ-BR23 items such as “breast symptoms” and “pain affected arm”; and BPI variables like “pain intensity” and “pain interference” (ρ = −0.222 to −0.446; *p* < 0.01 to 0.04). The results are presented in Figure 1.

The final regression model revealed that the variables “self-perceived muscle strength” from the IFIS and “nausea and vomiting” from the QLQ-C30 were significant predictors of higher self-perceived cardiorespiratory fitness. These factors collectively accounted for 47.2% of the variance in self-perceived cardiorespiratory fitness (adjusted r^2^ = 0.472; *p* < 0.01) in individuals who were at least 5 years post-cancer diagnosis. Additional details can be found in Table 4.

## 4. Discussion

This study aimed to explore the association between self-perceived cardiorespiratory fitness and health outcomes in LTBCSs and identify possible predictors in women at least 5 years post-breast-cancer diagnosis. The key findings indicate that five or more years after diagnosis, 35% of LTBCSs reported very poor/poor self-perceived cardiorespiratory fitness, 35% reported average levels, and 30% reported good/very good levels. Additionally, LTBCSs with lower self-perceived cardiorespiratory fitness levels demonstrated more significant declines in physical fitness, greater physical inactivity, increased CRF, higher pain levels, and poorer HRQoL. Notably, 47.2% of the variance in self-perceived cardiorespiratory fitness was explained by factors such as “self-perceived muscle strength” and “nausea and vomiting”.

Firstly, the findings from this study indicate that LTBCSs with poor self-perceived cardiorespiratory fitness exhibit significantly lower overall physical fitness than those with more favorable perceptions. Specifically, individuals in the very poor/poor category scored lower across all fitness domains than those in the good/very good and average groups. Similar trends have been observed in a study involving a mixed population of short- and long-term cancer survivors across various cancer type, where lower self-perceived fitness correlated with reduced functional performance and capacity [36]. As for the applicability to objective measures, two studies showed that higher self-reported fitness levels were associated with a higher objectively measured fitness level [5,37]. However, the lack of specificity regarding LTBCSs in these three previous studies limits direct comparison.

The PA levels also varied significantly based on self-perceived fitness, with a higher proportion of inactive LTBCSs in the very poor/poor category. This aligns with prior research showing that individuals perceiving themselves as fitter engage in more PA [38]. In BC survivors, self-efficacy and perceived physical abilities are key determinants of exercise adherence and structured program participation [39]. Given the benefits of cardiorespiratory fitness in cancer survivorship [40], interventions targeting self-perceived fitness—such as personalized programs focusing on endurance and symptom management—may enhance PA engagement and long-term health outcomes.

With respect to CRF, the analysis revealed significant differences in all PFS domains based on self-perceived cardiorespiratory fitness in LTBCSs. Specifically, those in the very poor/poor and average groups reported higher CRF levels across all domains compared to individuals in the good/very good group. This aligns with research linking lower self-perceived fitness to higher CRF levels and poorer well-being in cancer survivors [41]. However, no significant differences emerged between the very poor/poor and average groups, suggesting a stronger CRF impact in those perceiving their fitness as good/very good. While previous studies have linked objective cardiorespiratory fitness measures to CRF outcomes in BC survivors [42], evidence specifically addressing self-perceived cardiorespiratory fitness in LTBCSs remains limited. Given the clinical relevance of moderate-to-severe CRF in LTBCSs [43,44], these findings underscore the need to consider subjective fitness perceptions in CRF management strategies.

As for mood state, the analysis of the EVEA outcomes did not reveal significant differences between groups in any of the three comparisons. This suggests that self-perceived cardiorespiratory fitness did not have a significant impact on mood states among our LTBCSs. While previous studies in adolescents and short-term BC survivors have reported associations between higher self-perceived fitness and improved mood or lower levels of psychological distress [10,45,46,47,48], evidence within long-term BC survivorship remains scarce. Given these considerations, further research is needed to clarify the role of self-perceived cardiorespiratory fitness in emotional well-being among LTBCSs.

In relation to pain, LTBCSs with higher self-perceived cardiorespiratory fitness reported significantly lower pain intensity and interference, as measured by the BPI. This aligns with research showing that a home-based walking intervention improved cardiorespiratory fitness and reduced pain in patients undergoing treatment for solid tumors [49]. Although that study did not focus on self-perceived fitness, it suggested that PA may help alleviate pain. Similarly, studies in healthy adults linked better self-perceived fitness to lower pain sensitivity [50]. However, research on self-perceived cardiorespiratory fitness and pain in LTBCSs is scarce. Further studies are needed to clarify this relationship and its role in pain management.

In the case of HRQoL, LTBCSs with very poor/poor self-perceived cardiorespiratory fitness showed lower functioning, global health status, and summary scores, along with higher symptom burdens. While research on self-perceived cardiorespiratory fitness and HRQoL in LTBCSs is limited, studies in other populations suggest a strong link between perceived fitness and well-being. For instance, better self-perceived fitness in adolescents correlates with improved psychological well-being and lower distress [51]. In healthy adults, interventions targeting self-perceived fitness have been linked to physical and emotional health benefits [52]. Among survivors of BC, higher self-perceived physical fitness has been associated with better HRQoL and emotional functioning [10]. However, no studies have specifically examined self-perceived cardiorespiratory fitness and HRQoL in LTBCSs, highlighting the need for further research.

Considering the correlation and multiple regression analyses, our findings provide insights into the factors associated with self-perceived cardiorespiratory fitness in LTBCSs. Significant positive correlations were found between self-perceived cardiorespiratory fitness and various HRQoL functioning domains, as well as with overall self-perceived physical fitness. These findings align with previous research indicating that better perceptions of physical fitness are linked to enhanced HRQoL in survivors of BC [53]. Conversely, significant negative correlations were observed between self-perceived cardiorespiratory fitness and symptom-related variables, including the various dimensions of CRF, nausea and vomiting, dyspnea, insomnia, appetite loss, constipation, breast symptoms, and pain. These negative associations highlight the detrimental impact of symptom burden on perceived fitness levels, corroborating findings from prior studies that link higher symptom severity to reduced physical and emotional functioning in survivors of BC [36].

Finally, the regression analysis identified self-perceived muscle strength (IFIS) and nausea/vomiting (QLQ-C30) as significant predictors of self-perceived cardiorespiratory fitness, explaining 47.2% of the variance. Greater self-perceived muscle strength correlated with higher self-perceived cardiorespiratory fitness, while severe nausea and vomiting were linked to lower perceptions. Similar findings have been reported regarding self-perceived physical fitness and its impact on mental health in survivors of cancer [36]. However, the study [36] did not account for cancer type, treatment, or differences between short- and long-term survivors. Since symptom severity and patient needs vary by survivorship stage [54,55], further research is needed to assess the role of self-perceived cardiorespiratory fitness in LTBCSs.

Importantly, these findings suggest that interventions aimed at improving muscular strength and managing symptoms such as nausea could enhance self-perceived cardiorespiratory fitness in LTBCSs. Since higher self-perceived physical fitness has been associated with greater engagement in PA [10,56,57], strategies that target these aspects may encourage survivors to participate more actively in exercise programs. This could be particularly relevant in long-term survivorship, where maintaining PA is crucial for overall HRQoL [58].

This study has several limitations. The cut-off points used to classify participants align with previous categorizations [14,19], yet alternative values might have altered the results. Additionally, its cross-sectional design prevented establishing causal relationships between self-perceived cardiorespiratory fitness and physical, mental, and emotional variables, highlighting the need for longitudinal studies. The lack of prior research in LTBCSs using this analytical approach with the IFIS and these specific cut-off points further complicates comparisons. Future studies should build on these findings to explore self-perceived cardiorespiratory fitness in relation to health status and other fitness components, such as muscular strength, speed/agility, and flexibility. Moreover, anthropometric measurements (e.g., height, weight, and body composition) and objective cardiorespiratory fitness assessments were not included, as this study focused exclusively on self-perception. While this aligns with our primary aim, future research should consider incorporating these objective measures to better understand the relationship between self-perceived and actual fitness levels. This questions whether self-perceived results correspond to objective results or if participants under- or over-estimate their capabilities.

Despite its limitations, this study provides valuable insights into the association between self-perceived cardiorespiratory fitness and multiple health-related factors in LTBCSs. By identifying key determinants like self-perceived muscular strength and nausea/vomiting, it highlights the factors shaping the fitness perceptions in this population. The use of the IFIS, a validated tool for assessing perceived fitness across domains [13,14,15,59], enhances this study’s reliability, while the regression model explaining 47.2% of the variance strengthens clinical relevance. To the best of our knowledge, this is among the first studies to explore these relationships in LTBCSs, offering a novel perspective on the influence of self-perception on HRQoL. These findings not only advance scientific knowledge but also support the development of tailored therapeutic interventions and clinical guidelines.

## 5. Conclusions

In conclusion, only 30% of LTBCSs reported having good/very good levels of self-perceived cardiorespiratory fitness. Furthermore, LTBCSs with lower self-perceived cardiorespiratory fitness experienced a marked deterioration in fitness levels and increased sedentary behavior, as well as increased CRF, higher pain levels, and reduced HRQoL five or more years post-diagnosis. The combination of “self-perceived muscle strength” and “nausea and vomiting” explained 47.2% of the variance in self-perceived cardiorespiratory fitness among the LTBCSs.

These findings underscore the critical role of patients’ perceptions in the development of tailored rehabilitation strategies, emphasizing the necessity of integrating both subjective experiences and objective assessments to optimize long-term health outcomes and HRQoL in LTBCSs. From a practical standpoint, these results highlight the need for healthcare professionals to incorporate regular assessments of self-perceived fitness into survivorship care plans. Targeted interventions, such as personalized exercise programs and symptom management strategies, could help mitigate the negative effects of low self-perceived fitness and enhance overall well-being. Future research should explore the effectiveness of structured physical activity interventions in improving both perceived and objective fitness levels in this population.

## Figures and Tables

**Figure 1 healthcare-13-00718-f001:**
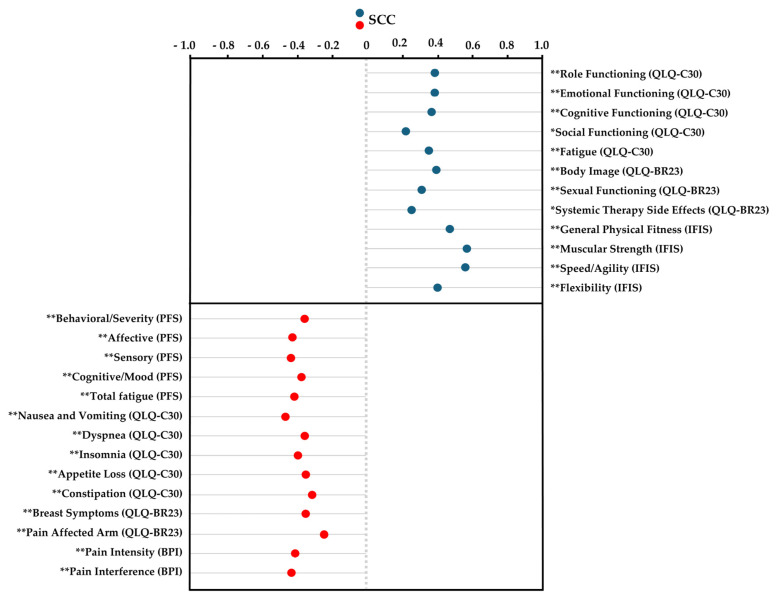
Spearman’s correlation coefficient for cardiorespiratory fitness using the International Fitness Scale (IFIS). Abbreviations: IFIS, International Fitness Scale; QLQ-C30, EORTC Core Quality of Life Quality of Life Questionnaire; QLQ-BR23, the breast-cancer-specific module; PFS, Piper Fatigue Scale; BPI, Brief Pain Inventory; SCC, Spearman’s correlation coefficient. * *p* ˂ 0.05; ** *p* < 0.01.

**Table 1 healthcare-13-00718-t001:** Demographic, clinical, and medical characteristics of LTBCSs according to the level of cardiorespiratory fitness.

	LTBCSs’ Cardiorespiratory Fitness			
Characteristics	Very Poor and Poor	Average	Good and Very Good	*p/x^2^*
	1–2 (IFIS)	3 (IFIS)	4–5 (IFIS)	
	(*n* = 28)	(*n* = 28)	(*n* = 24)	
Mean age ± SD (years)	47.20 ± 8.26	49.46 ± 7.09	52.04 ± 8.41	0.09 ^a^
Mean time since diagnosis ± SD (months)	90.15 ± 31.81	92.71 ± 28.16	88.13 ± 28.17	0.85 ^a^
Mean time since the first surgery ± SD (months)	86.53 ± 32.69	90.42 ± 28.23	85.52 ± 28.35	0.82 ^a^
Marital Status, *n* (%)				
Not married	5 (17.8)	7 (25)	1 (4.1)	0.06 ^b^
Married	17 (60.7)	19 (69.7)	19 (79.1)	
Divorced	6 (21.4)	2 (7.1)	0 (0)	
Widowed	0 (0)	0 (0)	4 (16.6)	
Educational level, *n* (%)				
Primary school	8 (28.5)	14 (50)	13 (54.1)	0.35 ^b^
Secondary school	9 (32.1)	5 (17.9)	4 (16.6)	
University	11 (39.2)	9 (32.1)	7 (29.1)	
Employment Status, *n* (%)				
Homemaker	10 (35.7)	5 (17.9)	11 (45.8)	0.28 ^b^
Currently working	5 (17.8)	7 (25)	5 (20.8)	
Sick leave	12 (42.8)	11 (39.3)	6 (25)	
Retired	1 (3.5)	5 (17.9)	2 (8.3)	
Tumor stage, *n* (%)				
I	5 (17.8)	6 (35.7)	4 (16.6)	0.53 ^b^
II	19 (6.8)	12 (50)	18 (75)	
IIIa	4 (14.2)	2 (14.3)	2 (8.3)	
Tumor Location, *n* (%)				
Right side	11 (39.2)	9 (32.1)	6 (25)	0.57 ^b^
Left side	15 (53.5)	17 (60.7)	18 (75)	
Bilateral	2 (7.1)	2 (7.1)	0 (0)	
Tobacco consumption, *n* (%)				
Non-smoker	13 (46.4)	13 (46.4)	14 (58.3)	0.15 ^b^
Smoker	10 (35.7)	5 (17.9)	3 (12.5)	
Ex-smoker	5 (17.8)	10 (35.7)	7 (29.1)	
Alcohol consumption, *n* (%)				
No consumption	10 (35.7)	9 (32.1)	11 (45.8)	0.06 ^b^
Monthly	7 (25)	11 (39.3)	2 (8.33)	
Weekly	8 (28.5)	7 (25)	11 (45.8)	
Daily	3 (10.7)	1 (13.6)	0 (0)	
Family history of breast cancer, *n* (%)				
No	11 (39.2)	16 (55.1)	13 (54.1)	0.32 ^b^
Yes	17 (60.7)	12 (42.9)	11 (45.8)	
Surgery, *n* (%)				
Lumpectomy	6 (21.4)	5 (17.9)	5 (20.8)	0.48 ^b^
Quadrantectomy	13 (46.4)	11 (39.3)	15 (62.5)	
Unilateral mastectomy	8 (28.5)	9 (32.1)	4 (16.6)	
Bilateral mastectomy	1 (3.5)	3 (10.7)	0 (0)	
Type of treatment, *n* (%)				
None	0 (0)	0 (0)	0 (0)	0.06 ^b^
Radiotherapy	1 (3.5)	2 (7.1)	0 (0)	
Chemotherapy	0 (0)	5 (17.9)	1 (4.1)	
Radiotherapy and chemotherapy	27 (96.4)	21 (75)	23 (95.8)	
Variety of medication, *n* (%)				
None	7 (25)	6 (21.4)	6 (25)	0.63 ^b^
Tamoxifen	13 (46.4)	9 (32.1)	8 (33.3)	
Other types	8 (28.5)	13 (46.4)	10 (41.6)	
Metastasis, *n* (%)				
No	21 (75)	24 (85.7)	21 (87.5)	0.49 ^b^
Yes	7 (25)	4 (14.3)	3 (12.5)	
Recurrence, *n* (%)				
No	24 (85.7)	22 (78.6)	21 (87.5)	0.65 ^b^
Yes	4 (14.2)	6 (21.4)	3 (12.5)	
Menopause, *n* (%)				
No	3 (10.7)	3 (10.7)	5 (20.8)	0.41 ^b^
Yes	25 (89.2)	25 (89.3)	19 (79.1)	
Currently seeing a psychologist or in the last three months, *n* (%)				
No	12 (42.8)	10 (35.7)	11 (45.8)	0.68 ^b^
Yes	16 (57.1)	18 (64.3)	13 (54.1)	
Currently seeing a physiotherapist or in the last three months, *n* (%)				
No	11 (39.2)	10 (35.7)	9 (37.5)	0.97 ^b^
Yes	17 (60.7)	18 (64.3)	15 (62.5)	

Abbreviations: LTBCSs, long-term breast cancer survivors; IFIS, International Fitness Scale; n, sample size; SD, standard deviation. p values for between-group differences were calculated using the ANOVA test ^a^ and X^2^ for categorical variables ^b^.

**Table 2 healthcare-13-00718-t002:** Physical fitness, physical activity level, cancer-related fatigue, mood state, and pain of LTBCSs according to the level of cardiorespiratory fitness.

Variables		LTBCSs Cardiorespiratory Fitness								

		Very Poor/Poor	Average	Good/Very Good						
		1–2	3	4–5						
		(IFIS)	(IFIS)	(IFIS)	*p* Values	Cohen’s * d*	*p* Values	Cohen’s * d*	*p* Values	Cohen’s * d*
		(*n* = 28)	(*n* = 28)	(*n* = 24)	1–2 vs. 3	1–2 vs. 3	1–2 vs. 4–5	1–2 vs. 4–5	3 vs. 4–5	3 vs. 4–5
IFIS, mean ± SD (95% CI) ^a^										
General physical fitness		2.79 ± 0.94 (2.43–3.15)	3.25 ± 0.79 (2.94–3.55)	3.95 ± 0.92 (3.55–4.35)	0.07	0.53	<0.01 **	>1.20	<0.01 **	0.81
Muscular strength		2.34 ± 0.89 (2.00–2.68)	2.75 ± 0.84 (2.42–3.07)	3.69 ± 0.87 (3.31–4.07)	0.06	0.47	<0.01 **	>1.20	<0.01 **	1.10
Speed/agility		2.37 ± 0.82 (2.06–2.69)	2.89 ± 0.68 (2.62–3.15)	3.73 ± 0.91 (3.34–4.13)	0.01 *	0.69	<0.01 **	>1.20	<0.01 **	1.05
Flexibility		2.44 ± 0.98 (2.07–2.82)	2.71 ± 0.97 (2.33–3.09)	3.60 ± 0.78 (3.27–3.94)	0.31	0.28	<0.01 **	>1.20	<0.01 **	1.01
MLTPA (MET hour/week), *n* (%) ^b^										
Inactive (≤ 3)		12 (42.8)	4 (14.3)	5 (20.8)	0.07	-	0.02 *	-	0.53	-
Low activity (3.1–7.4)		9 (32.1)	14 (50)	8 (33.3)						
Active (≥ 7.5)		7 (25)	10 (35.7)	11 (45.8)						
PFS domains, mean ± SD (95% CI) ^a^										
Behavioral/severity		3.71 ± 3.10 (2.53–4.89)	3.77 ± 2.86 (2.66–4.88)	1.13 ± 1.43 (0.51–1.75)	0.94	0.02	<0.01 **	1.07	0.02 *	1.17
Affective		4.21 ± 3.29 (2.95–5.46)	4.05 ± 13.09 (2.85–5.25)	1.30 ± 2.14 (0.37–2.23)	0.70	0.02	<0.01 **	1.05	<0.01 **	0.29
Sensory		4.31 ± 3.15 (3.11–5.51)	4.26 ± 3.00 (3.09–5.43)	1.45 ± 2.08 (0.55–2.35)	0.86	0.02	<0.01 **	1.07	<0.01 **	1.09
Cognitive/mood		3.91 ± 3.12 (2.73–5.10)	3.96 ± 2.89 (2.84–5.08)	1.37 ± 1.93 (0.54–2.21)	0.94	0.02	<0.01 **	0.98	<0.01 **	1.05
Total fatigue score		4.02 ± 2.92 (2.90–5.13)	4.04 ± 2.77 (2.96–5.11)	1.31 ± 1.67 (0.58–2.03)	0.89	0.01	<0.01 **	1.14	<0.01 **	1.19
PFS (Cut score type A), (%) ^b^										
No fatigue	0–0.9	6 (21.4)	6 (21.4)	14 (58.3)	0.66	-	0.03 *	-	<0.01 **	-
Mild	1–3.9	5 (17.8)	8 (28.6)	8 (33.3)						
Moderate	4–6.9	13 (46.4)	9 (32.1)	2 (8.3)						
Severe	7–10	4 (14.2)	5 (17.9)	0 (0)						
PFS (Cut score type B), (%) ^b^										
No fatigue	0–0.9	6 (21.4)	6 (21.4)	14 (58.3)	0.90	-	0.01*	-	<0.01 **	-
Mild	1–2.9	5 (17.8)	5 (17.9)	6 (25)						
Moderate	3–5.9	9 (32.1)	11 (39.3)	4 (16.6)						
Severe	6–10	8 (28.5)	6 (21.4)	0 (0)						
EVEA, mean ± SD (95% CI) ^a^										
Sadness–depression		3.45 ± 2.79 (2.39–4.51)	3.08 ± 2.92 (1.95–4.22)	2.03 ± 2.15 (1.10–2.96)	0.58	0.13	0.22	0.57	0.32	0.41
Anxiety		3.32 ± 2.81 (2.25–4.39)	3.53 ± 2.92 (2.40–4.66)	2.13 ± 1.63 (1.42–2.83)	0.79	0.07	0.29	0.52	0.15	0.59
Anger–hostility		2.62 ± 2.86 (1.53–3.70)	2.52 ± 2.60 (1.51–3.53)	1.44 ± 1.60 (0.75–2.14)	0.89	0.04	0.45	0.51	0.21	0.50
Happiness		6.45 ± 9.61 (2.79–10.11)	5.33 ± 2.15 (4.50–6.17)	6.20 ± 2.57 (5.09–7.31)	0.43	0.16	0.15	0.04	0.20	0.37
VAS (cm), mean ± SD (95% CI) ^a^										
Affected arm		2.79 ± 2.87 (1.70–3.88)	2.57 ± 2.63 (1.55–3.59)	1.39 ± 1.94 (0.54–2.23)	0.64	0.08	0.08	0.57	0.09	0.51
Unaffected arm		1.75 ± 2.98 (0.62–2.89)	1.71 ± 2.89 (0.59–2.83)	0.78 ± 1.88 (−0.03–1.5)	0.98	0.01	0.21	0.39	0.10	0.38
BPI, mean ± SD (95% CI) ^a^										
Intensity		3.31 ± 2.76 (2.25–4.36)	2.21 ± 2.17 (1.37–3.05)	1.05 ± 1.97 (0.20–1.91)	0.15	0.44	<0.01 **	0.94	0.02 *	0.56
Interference		3.18 ± 2.95 (2.06–4.30)	2.01 ± 2.59 (1.00–3.02)	0.58 ± 1.47 (−0.05–1.21)	0.12	0.42	<0.01 **	1.12	0.02 *	0.68

Abbreviations: LTBCSs, long-term breast cancer survivors; IFIS, International Fitness Scale; MLTPA, Minnesota Leisure Time Physical Activity; MET, metabolic equivalent task; PFS, Piper Fatigue Scale; EVEA, Scale for Mood Assessment; VAS, Visual Analog Scale; BPI, Brief Pain Inventory; CI, confidence interval; n, sample size; SD, standard deviation; Note: The row corresponding to cardiorespiratory fitness (IFIS) was not included in the table as it was the independent grouping variable. *p* values for between-group differences were calculated using the Mann–Whitney U test ^a^ for non-normal and chi-square ^b^ test for categorical variables. Between-group effect sizes were calculated using Cohen’s *d* for continuous variables ^a^. * *p* ˂ 0.05; ** *p* < 0.01.

**Table 3 healthcare-13-00718-t003:** Health-related quality of life of LTBCSs according to their cardiorespiratory fitness.

	LTBCSs Cardiorespiratory Fitness								
Variables	Very Poor and Poor	Average	Good and Very Good						
	1–2	3	4–5						
	(IFIS)	(IFIS)	(IFIS)	*p* values	Cohen’s *d*	*p* Values	Cohen’s *d*	*p* Values	Cohen’s *d*
	(*n* = 28)	(*n* = 28)	(*n* = 24)	1–2 vs. 3	1–2 vs. 3	1–2 vs. 4–5	1–2 vs. 4–5	3 vs. 4–5	3 vs. 4–5
Functioning Scales QLQ-C30, mean ± SD (95% CI)									
Physical Functioning	32.18 ± 18.86 (25.00–39.35)	35.10 ± 24.14 (25.74–44.46)	22.98 ± 11.47 (24.01–33.94)	0.83	0.13	0.82	0.59	0.55	0.64
Role functioning	72.38 ± 25.77 (62.57–82.18)	83.45 ± 14.38 (77.87–89.03)	90.97 ± 13.57 (85.10–96.84)	0.15	0.53	<0.01 **	0.90	0.02 *	0.54
Emotional functioning	66.66 ± 34.78 (53.43–79.89)	83.33 ± 23.56 (74.19–92.47)	90.57 ± 22.37 (80.90–100.25)	0.05 *	0.56	<0.01 **	0.82	0.13	0.32
Cognitive functioning	54.59 ± 30.42 (43.02–66.16)	62.49 ± 27.91 (51.67–73.32)	81.15 ± 29.00 (68.61–93.69)	0.36	0.27	<0.01 **	0.89	<0.01 **	0.66
Social functioning	56.89 ± 27.28 (46.51–67.27)	57.73 ± 36.42 (43.61–71.86)	73.18 ± 25.98 (61.95–84.42)	0.74	0.03	0.11	0.61	0.15	0.49
Symptom Scales QLQ-C30, mean ± SD (95% CI)									
Fatigue	82.60 ± 31.17 (69.12–96.08)	75.59 ± 29.21 (64.26–86.92)	61.49 ± 32.76 (49.03–73.95)	0.09	0.23	0.02 *	0.66	0.27	0.45
Nausea and Vomiting	51.72 ± 32.83 (39.23–64.21)	34.12 ± 27.70 (23.38–44.86)	20.29 ± 24.76 (9.58–31.00)	0.04 *	0.58	<0.01 **	1.08	0.05 *	0.53
Pain	8.62 ± 17.03 (2.14–15.09)	10.71 ± 24.09 (1.37–20.05)	2.17 ± 7.62 (−1.12–5.47)	0.89	0.10	0.29	0.49	0.17	0.48
Single Items QLQ-C30, mean ± SD (95% CI)									
Dyspnea	50.57 ± 33.17 (37.95–63.19)	37.49 ± 29.61 (26.01–48.98)	25.36 ± 29.67 (12.53–38.19)	0.13	0.42	0.01 *	0.80	0.10	0.41
Insomnia	35.63 ± 35.55 (22.10–49.15)	21.42 ± 27.53 (10.74–32.10)	10.14 ± 23.42 (0.01–20.27)	0.12	0.45	<0.01 **	0.85	0.06	0.44
Appetite loss	60.34 ± 35.19 (46.95–73.73)	52.38 ± 31.98 (39.97–64.78)	34.78 ± 30.94 (21.40–48.16)	0.33	0.24	0.02 *	0.77	0.06	0.56
Constipation	22.98 ± 35.74 (9.38–36.58)	7.14 ± 16.62 (0.69–13.58)	2.89 ± 9.60 (−1.25–7.05)	0.06	0.57	0.02 *	0.77	0.33	0.31
Diarrhea	28.73 ± 37.50 (14.46–43.00)	29.16 ± 31.95 (16.77–41.55)	15.94 ± 22.17 (6.35–25.53)	0.76	0.01	0.37	0.42	0.14	0.48
Financial Difficulties	9.19 ± 23.39 (0.29–18.09)	16.66 ± 26.44 (6.41–26.92)	10.14 ± 18.62 (2.08–18.19)	0.13	0.30	0.31	0.04	0.40	0.29
Global Health Status QLQ-C30, mean ± SD (95% CI)									
Global health status	10.28 ± 25.44 (−0.71–21.29)	22.98 ± 34.62 (9.81–36.15)	33.33 ± 36.28 (19.26–47.40)	0.19	0.42	0.03 *	0.74	<0.01 **	0.29
Summary Score QLQ-C30, mean ± SD (95% CI)									
Summary score	58.66 ± 17.02 (52.18–65.13)	65.69 ± 13.13 (60.60–70.79)	74.66 ± 11.65 (69.62–79.70)	0.12	0.46	<0.01 **	1.10	<0.01 **	0.72
Functional Scales QLQ-BR23, mean ± SD (95% CI)									
Body image	54.31 ± 23.21 (45.48–63.13)	62.60 ± 18.76 (54.92–69.48)	77.17 ± 19.97 (68.53–85.81)	0.15	0.39	<0.01 **	1.06	0.01 *	0.75
Sexual functioning	69.82 ± 30.97 (58.04–81.60)	76.78 ± 26.19 (66.63–86.94)	86.95 ± 24.20 (76.48–97.42)	0.35	0.24	0.01 *	0.62	0.04 *	0.40
Sexual enjoyment	16.66 ± 18.89 (9.47–23.85)	26.78 ± 21.43 (18.47–35.09)	22.46 ± 22.81 (12.59–32.32)	0.05 *	0.50	0.18	0.28	0.49	0.20
Future perspective	32.18 ± 25.94 (22.31–42.05)	36.90 ± 22.84 (28.04–45.75)	26.08 ± 26.50 (14.62–37.54)	0.46	0.19	0.21	0.23	0.06	0.44
Symptom Scales QLQ-BR23, mean ± SD (95% CI)									
Systemic therapy side effects	47.12 ± 38.33 (32.54–61.70)	47.61 ± 36.77 (33.36–61.87)	72.46 ± 35.74 (57.00–87.92)	0.96	0.01	0.02 *	0.68	0.01 *	0.69
Breast symptoms	32.51 ± 23.40 (23.61–41.41)	33.50 ± 18.84 (26.19–40.81)	17.75 ± 22.46 (8.03–27.46)	0.61	0.05	<0.01 **	0.64	<0.01 **	0.76
Arm symptoms	30.74 ± 30.62 (19.09–42.39)	28.57 ± 25.19 (18.80–38.34)	17.39 ± 24.60 (6.75–28.03)	0.94	0.08	0.11	0.48	0.05 *	0.45
Upset by hair loss	33.33 ± 30.42 (21.75–44.90)	37.30 ± 33.01 (24.49–25.10)	23.18 ± 28.01 (11.07–35.30)	0.70	0.13	0.21	0.35	0.09	0.46

Abbreviations: LTBCSs, long-term breast cancer survivors; QLQ-C30, EORTC Core Quality of Life Quality of Life Questionnaire; QLQ-BR23; the breast-cancer-specific module; CI, confidence interval; n, sample size; SD, standard deviation. p values for between-group differences were calculated using the Mann–Whitney U test. Between-group effect sizes were calculated using Cohen’s *d* for continuous variables. * *p* ˂ 0.05; ** *p* < 0.01.

**Table 4 healthcare-13-00718-t004:** Summary of stepwise multiple regression analysis to determine predictors of self-perceived cardiorespiratory fitness using the International Fitness Scale (IFIS).

	Model 1 r^2^ = 0.379						Model 2 r^2^ = 0.472				
Variables /Predictors	β	95% CI	*t*	*p*	Linear Regression Equation Y = a + bX	Variables /Predictors	β	95% CI	*t*	*p*	Linear Regression Equation Y = a + bX
Self-perceived muscle strength (IFIS)	0.61	0.47 ± 0.85	6.90	<0.01 **	Self-perceived cardiorespiratory fitness = 0.98 + (0.66 Self-perceived muscle strength)	Self-perceived muscle strength (IFIS)	0.40	0.22 ± 0.65	4.01	<0.01 **	Self-perceived cardiorespiratory fitness = 0.38 + (0.43 Self-perceived muscle strength) + (−0.42 Nausea and vomiting)
						Nausea and vomiting (QLQ-C30)	−0.37	−0.19 ± −0.65	−3.67	<0.01 **	

Dependent variable: Self-perceived cardiorespiratory fitness; r^2^, adjusted coefficient of determination; β, regression coefficient; t, coefficient t-value. Abbreviations: IFIS, International Fitness Scale; QLQ-C30, EORTC Core Quality of Life Quality of Life Questionnaire; CI, confidence interval; ** *p* < 0.01.

## Data Availability

The datasets generated during and/or analyzed during the current study are available from the corresponding author upon reasonable request.

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
