# Peer review of "Exploring Predictors of Self-Perceived Cardiorespiratory Fitness ≥ 5 Years Beyond Breast Cancer Diagnosis: A Cross-Sectional Study"

_healthcare, 2025, doi:10.3390/healthcare13070718_

Round 1
Reviewer 1 Report
Comments and Suggestions for Authors
First of all, I have to say that the work has a great applicability and can help both researchers and patients to improve the quality of life of people with breast cancer.
However, there are some points to be taken into account.
First of all, the materials and methods, the first paragraph is the same, and the sample is also the same as another study published by you entitled: ‘Assessing the Relationship of Different Levels of Pain to the Health Status of Long-Term Breast Cancer Survivors: A Cross-Sectional Study’. Be careful with this, as using the same sample and similar data for different papers can be exclusionary conditions of the study.
Both the abstract and the introduction are complete and clear.
The methodology is quite complete but I have some questions that would be clarifying for readers or researchers who want to replicate this study. How the questionnaires were passed out, all on the same day, on different days. Who was the person specialised in passing out the questionnaires. Were they evaluated by the same person?
Please, when putting tables in the document, check the APA guidelines for tables, as it will make the document much cleaner and much better understood by the readers.
When you talk about the results, you are giving a summary of the results and then you do give a conclusion. However, I recommend that you expand that conclusion and also talk about the practical applications of this research, so that it has greater visibility and is of great use to other professionals in the field.
Finally, when using abbreviations at the end of the document, when the abbreviation appears for the first time in the document, try to write what it means in full and then in brackets what the abbreviation means, so that you can then use the abbreviation throughout the text.
Author Response
Maria Figueroa Mayordomo
Department of Physiotherapy, Faculty of Health Sciences,
European University of Valencia,
46112 Valencia, Spain
Tel +34-663746655
Editorial Reviewer 1
Healthcare
7 March 2025
Dear Reviewer 1,
First of all, I have to say that the work has a great applicability and can help both researchers and patients to improve the quality of life of people with breast cancer. However, there are some points to be taken into account.
Author response:
First of all, we would like to thank you for your words and the time dedicated to the understanding and improvement of this scientific work. In this way, and from here on, all the answers are detailed, individually and by sections, to each of your suggestions or comments.
Please find below the answers to each of your contributions:
Materials and methods
First of all, the materials and methods, the first paragraph is the same, and the sample is also the same as another study published by you entitled: ‘Assessing the Relationship of Different Levels of Pain to the Health Status of Long-Term Breast Cancer Survivors: A Cross-Sectional Study’. Be careful with this, as using the same sample and similar data for different papers can be exclusionary conditions of the study.
Author response:
Thank you very much for your valuable observation. While both studies were conducted at iMUDS and had an estimated sample size of 80 LTBCSs, they did not consist of the exact same individuals. The recruitment process was independent, and the inclusion criteria allowed for participant variability between studies. Additionally, the study periods differed, with the previous study being conducted in 2022 and the current study spanning late 2022 to early 2023. This distinction ensures that the data are unique to each study. To enhance clarity and avoid misunderstandings, we have revised the Methods section to better reflect the specific characteristics of the sample in this study.
Similarly, to prevent any potential confusion, we have thoroughly reviewed this information and revised the text accordingly. (Please refer to the beginning of the Materials and Methods section to verify these changes. Lines 108 to 153).
“Design and participants
This cross-sectional study was conducted between late 2022 and early 2023 at the Sport and Health Joint University Institute (iMUDS), following the Declaration of Helsinki (14/2017) [18] and approved by the Biomedical Research Ethics Committee of Granada (CEIm) (1038-N-16 I.P/07/26/2018).
A total of 80 LTBCSs participated in this study. Eligible participants were identified and recruited through the oncology department of the University Hospital Complex of Granada during the specified study period.
Participants were informed about the study objectives via telephone and given the opportunity to ask questions before providing written informed consent during an in-person assessment session conducted by a trained physiotherapist from the research team. Each session lasted approximately one hour.
Eligibility criteria and group classification
Inclusion criteria:
- Female sex.
- Age ≥ 18 years.
- Diagnosis of stage I–IIIa breast cancer at least five years prior to enrollment.
- No participation in structured fitness-related assessments within the past three months (to avoid bias in self-reported fitness levels).
Exclusion criteria:
- Any condition preventing the completion of the assessment protocol.
- Inability to understand the study procedures.
Participants were informed about the study objectives via telephone and given the opportunity to ask questions before providing written informed consent during an in-person assessment session conducted by a trained physiotherapist from the research team. Each session lasted approximately one hour and took place on a single day, during which participants completed a comprehensive assessment dossier. This dossier included all questions related to demographic and clinical variables, as well as validated questionnaires assessing the study variables.
A trained physiotherapist, experienced in oncology patient assessment, supervised the process, ensuring participants understood the questionnaires and providing assistance if needed. The physiotherapist overseeing the assessment was different from the team member responsible for digitizing the responses, ensuring data integrity. Additionally, the statistical analysis was conducted by a separate researcher, independent from both the assessment and data entry processes.
The necessary sample size was estimated using G*Power (Version 3.1.9.7) for a comparison of three independent groups. Assuming a medium effect size (f = 0.25), an alpha level of 0.05, and a power of 0.80, the required total sample size was 72 participants (24 per group). Finally, we recruited 80 participants, ensuring adequate statistical power for the analyses”.
Abstract and introduction
Both the abstract and the introduction are complete and clear.
Author response:
Thank you for your positive feedback. We are glad to hear that both the abstract and the introduction are clear and complete. We appreciate your thorough review and constructive comments, which have helped us improve the manuscript.
Methodology
The methodology is quite complete but I have some questions that would be clarifying for readers or researchers who want to replicate this study. How the questionnaires were passed out, all on the same day, on different days. Who was the person specialized in passing out the questionnaires. Were they evaluated by the same person?
Author response:
Thank you very much for your contribution. We believe you are absolutely right. However, when we addressed your first suggestion, we realized that this comment was also along the same lines. Therefore, once the first comment was resolved, the same changes were applied here as well (Please see lines 118 to 131).
“Participants were informed about the study objectives via telephone and given the opportunity to ask questions before providing written informed consent during an in-person assessment session conducted by a trained physiotherapist from the research team. Each session lasted approximately one hour and took place on a single day, during which participants completed a comprehensive assessment dossier. This dossier included all questions related to demographic and clinical variables, as well as validated questionnaires assessing the study variables.
A trained physiotherapist, experienced in oncology patient assessment, supervised the process, ensuring participants understood the questionnaires and providing assistance if needed. The physiotherapist overseeing the assessment was different from the team member responsible for digitizing the responses, ensuring data integrity. Additionally, the statistical analysis was conducted by a separate researcher, independent from both the assessment and data entry processes”.
Tables
Please, when putting tables in the document, check the APA guidelines for tables, as it will make the document much cleaner and much better understood by the readers.
Author response:
Thank you very much for your appreciation once more. We have taken your suggestion into account and have revised all the tables and graphs to align with your proposal. We believe they are now easier to understand and adhere to the recommended guidelines.
Results
When you talk about the results, you are giving a summary of the results and then you do give a conclusion. However, I recommend that you expand that conclusion and also talk about the practical applications of this research, so that it has greater visibility and is of great use to other professionals in the field.
Author response:
Thank you very much for your contribution. However, we don’t completely understand which conclusion section you are referring to when mentioning the results section. We have presented the results section as objectively as possible, so as to not make any interpretation of the data before discussing it with available literature.
Furthermore, as there is no conclusion in the results, we think it might be possible that you meant the way the results are presented in the discussion and the final conclusion. Therefore, we have deemed it appropriate to attach the different paragraphs of the discussion where the clinical implications of our results are detailed as each variable has been analyzed (see below). Additionally, we reformulated the conclusion to add a new paragraph that further explores the clinical implications of this research (Please refer to the discussion and conclusion sections of the article to see these changes).
Discussion:
- Given the benefits of cardiorespiratory fitness in cancer survivorship [39], interventions targeting self-perceived fitness—such as personalized programs focusing on endurance and symptom management—may enhance PA engagement and long-term health outcomes (lines 423 to 426).
- Given the clinical relevance of moderate-to-severe CRF in LTBCSs [42,43], these findings underscore the need to consider subjective fitness perceptions in CRF management strategies (lines 436 to 438).
- Given these considerations, further research is needed to clarify the role of self-perceived cardiorespiratory fitness in emotional well-being among LTBCSs (lines 445 to 446).
- However, research on self-perceived cardiorespiratory fitness and pain in LTBCSs is scarce. Further studies are needed to clarify this relationship and its role in pain management (lines 453 – 455).
- However, no studies have specifically examined self-perceived cardiorespiratory fitness and HRQoL in LTBCSs, highlighting the need for further research (lines 464 to 466).
- Similar findings have been reported regarding self-perceived physical fitness and its impact on mental health in cancer survivors [36]. However, said study [36] did not account for cancer type, treatment, or differences between short- and long-term survivors. Since symptom severity and patient needs vary by survivorship stage [53,54], further research is needed to assess the role of self-perceived cardiorespiratory fitness in LTBCSs (lines 484 to 489).
- Since higher self-perceived physical fitness has been associated with greater engagement in PA [10,55,56], strategies that target these aspects may encourage survivors to participate more actively in exercise programs. This could be particularly relevant in long-term survivorship, where maintaining PA is crucial for overall HRQoL [57] (lines 492 to 496).
- The lack of prior research in LTBCSs using this analytical approach with the IFIS and these specific cut-off points further complicates comparisons. Future studies should build on these findings to explore self-perceived cardiorespiratory fitness in relation to health status and other fitness components, such as muscular strength, speed/agility, and flexibility (lines 501 to 505).
- To our knowledge, this is among the first studies to explore these relationships in LTBCSs, offering a novel perspective on the influence of self-perception on HRQoL. These findings not only advance scientific knowledge but also support the development of tailored therapeutic interventions and clinical guidelines (lines 518 to 521).
Conclusion:
- In conclusion, only 30% of LTBCSs reported having good/very good levels of self-perceived cardiorespiratory fitness. Furthermore, LTBCSs with lower self-perceived cardiorespiratory fitness experienced a marked deterioration in fitness levels and increased sedentary behavior, as well as increased CRF, higher pain levels, and reduced HRQoL five or more years post-diagnosis. The combination of "self-perceived muscle strength" and "nausea and vomiting" explained 47.2% of the variance in self-perceived cardiorespiratory fitness among LTBCSs.
These findings underscore the critical role of patients’ perceptions in the development of tailored rehabilitation strategies, emphasizing the necessity of integrating both subjective experiences and objective assessments to optimize long-term health outcomes and HRQoL in LTBCSs. From a practical standpoint, these results highlight the need for healthcare professionals to incorporate regular assessments of self-perceived fitness into survivorship care plans. Targeted interventions, such as personalized exercise programs and symptom management strategies, could help mitigate the negative effects of low self-perceived fitness and enhance overall well-being. Future research should explore the effectiveness of structured physical activity interventions in improving both perceived and objective fitness levels in this population (lines 523 – 539).
Abbreviations
Finally, when using abbreviations at the end of the document, when the abbreviation appears for the first time in the document, try to write what it means in full and then in brackets what the abbreviation means, so that you can then use the abbreviation throughout the text.
Author response:
Thank you very much for your suggestion. We have reviewed the entire document to ensure this premise is met, and we believe that everything is now correctly adapted.
The author responsible for correspondence is:
Maria Figueroa Mayordomo
Department of Physiotherapy, Faculty of Health Sciences,
European University of Valencia,
46112 Valencia, Spain
Tel +34-663746655
Sincerely,
Maria Figueroa Mayordomos

Reviewer 2 Report
Comments and Suggestions for Authors
The manuscript entitled “Exploring predictors of self-perceived cardiorespiratory fitness ≥ 5 years beyond breast cancer diagnosis: a cross-sectional study” was evaluated. The study provides important information for the scientific community; however, adjustments need to be made before it can be suggested for publication.
Below are some notes and suggestions that should be worked on in the text by the authors. I kindly ask that the changes made to the manuscript be highlighted in red or highlighted in yellow.
Abstract
1- In lines 25, 28, 33, 39 and 40, what does the acronym “LTBCSs” mean? Please include the full name before the first abbreviation.
2- The objective of the study is too long and redundant. Please adjust it to make it clearer in relation to the purpose of the work. Also adjust the objective at the end of the introduction.
3- In the methodology of the abstract, please include whether anthropometric and body composition assessments were performed on the participants, as well as their sociodemographic profile.
4- In the results of the abstract, include information about the “p” values in relation to the statistical analyses performed. If possible, also mention which main statistical analyses were performed in the methodology.
5- In the conclusion of the abstract, you are talking about the results of the study. I suggest reviewing the conclusion focusing on the objectives and that the information about the results be included in the results topic.
Keywords
6- Avoid repeating the same words that were presented in the title, the function of keywords is to expand the search for the study.
Introduction
7- In the introduction, you advocate the self-perceived assessment of the health components of women who have had breast cancer for more than 5 years. However, what I missed in the introduction was seeing results from previous clinical studies that made this assessment with people who had cancer or with other special groups, as well as associating or correlating self-perceived health with physical tests to see if they really follow the same path. Therefore, I believe that including this type of study in the introduction could make the introduction more robust.
Materials and Methods
8- I could not find the clinical trial registration, please insert the registration number and the website link.
9- Please include the study flowchart in the text. Follow the CONSORT 2010 model (https://view.officeapps.live.com/op/view.aspx?src=https%3A%2F%2Fwww.equator-network.org%2Fwp-content%2Fuploads%2F2013%2F09%2FCONSORT-2010-Flow-Diagram-MS-Word.doc&wdOrigin=BROWSELINK).
10- In the methodology, I did not find any information about anthropometric and body composition assessments of the participants, nor did I find any assessment of physical tests to compare with self-reported health information. If this was done, please include it, as these factors affect the perception of health and quality of life, as well as physical fitness. Describe how they were carried out, as well as the brands and scales of the equipment.
11- I am unsure about how the recruitment of people was carried out so that you could distribute them into the three groups. Was it convenient? That is why it is important to present the research flowchart.
Results
12- Figure 1 is not in very good resolution, please improve the image quality and font size.
13- After figure 1, an untitled frame appears, please adjust it. I believe it could be table 4, however, it was created in frame format.
Discussion
14- The discussion should be strengthened with more clinical studies that evaluated these same variables and included studies with people who performed physical tests to prove whether perception can also be portrayed in people's assessments.
15- Regarding the limitations of the study, I believe that a major limitation is the fact that you did not perform physical tests on people. This prevents us from knowing whether people were reporting what their bodies could respond to when challenged in a physical test. This question raises the question: if these people were previously subjected to physical tests to assess strength, muscular endurance, flexibility and cardiorespiratory capacity, would they have answered the questionnaire in the same way?
Conclusions
16- Please try to make the conclusion more direct in relation to the objectives.
References
17- It has been found that there are many references that are more than 5 years old, please try to replace them with more recent references.
Author Response
Maria Figueroa Mayordomo
Department of Physiotherapy, Faculty of Health Sciences,
European University of Valencia,
46112 Valencia, Spain
Tel +34-663746655
Editorial Reviewer 2
Healthcare
7 March 2025
Dear Reviewer 2,
The manuscript entitled “Exploring predictors of self-perceived cardiorespiratory fitness ≥ 5 years beyond breast cancer diagnosis: a cross-sectional study” was evaluated. The study provides important information for the scientific community; however, adjustments need to be made before it can be suggested for publication.
Below are some notes and suggestions that should be worked on in the text by the authors. I kindly ask that the changes made to the manuscript be highlighted in red or highlighted in yellow.
Author response:
First of all, we would like to thank you for your words and the time dedicated to the understanding and improvement of this scientific work. In this way, and from here on, all the answers are detailed, individually and by sections, to each of your suggestions or comments.
Please find below the answers to each of your contributions:
Abstract
- In lines 25, 28, 33, 39 and 40, what does the acronym “LTBCSs” mean? Please include the full name before the first abbreviation.
Author response:
Thank you very much for your contribution. It is absolutely true; due to an oversight, we did not realize that we had not mentioned the meaning of the abbreviations. We have now reviewed the entire document to ensure that both these abbreviations and the others are properly introduced. (Please refer to lines 24 to 27 to observe this change).
“Abstract: Background/Objectives: This study aimed to examine the relationship between self-perceived cardiorespiratory fitness levels and health outcomes among long-term breast cancer survivors (LTBCSs), as well as identify predictors of self-perceived cardiorespiratory fitness in women with at least 5 years post-cancer diagnosis”.
- The objective of the study is too long and redundant. Please adjust it to make it clearer in relation to the purpose of the work. Also adjust the objective at the end of the introduction.
Author response:
Thank you very much for your contribution. Taking your consideration into account, we have decided to modify the objective in the abstract (lines 24 to 27), at the end of the introduction (lines 104 to 106), and at the beginning of the discussion (lines 398 to 400) to ensure greater coherence and uniformity in expressing the scientific content of this research.
Abstract:
This study aimed to examine the association between self-perceived cardiorespiratory fitness and health outcomes in long-term breast cancer survivors (LTBCSs) and identify possible predictors in women at least 5 years post-diagnosis.
Introduction:
Therefore, the purpose of this study was to explore the association between self-perceived cardiorespiratory fitness and health outcomes in LTBCSs and identifies possible predictors in women at least 5 years post-breast cancer diagnosis.
Discussion:
This study aimed to explore the association between self-perceived cardiorespiratory fitness and health outcomes in LTBCSs and identifies key predictors in women at least 5 years post-breast cancer diagnosis.
- In the methodology of the abstract, please include whether anthropometric and body composition assessments were performed on the participants, as well as their sociodemographic profile.
Author response:
Thank you for your insightful suggestion. We acknowledge your comment regarding the methodology section in the abstract. To address your concern, we have modified the abstract to include that sociodemographic and clinical data were also collected. However, we would like to clarify that our study did not include anthropometric or body composition assessments (e.g., height, weight, or body fat percentage) (Please refer to lines 27 to 34 to see the revised text).
“Methods: A cross-sectional study was carried out involving 80 LTBCSs, divided into three groups according to their self-reported cardiorespiratory fitness levels: very poor/poor (1–2), average (3), and good/very good (4–5). Sociodemographic and clinical data were collected, and the study analyzed variables measured at least five years after diagnosis, focusing on various factors including physical fitness, physical activity (PA) levels, cancer-related fatigue (CRF), mood, pain, and health-related quality of life (HRQoL). ANOVA, Mann-Whitney U, and Chi-square tests were performed, along with correlation and multiple regression analysis. Cohen's d was used to calculate effect sizes”.
Additionally, we have attached the first paragraph of the variables section, where the type of variables considered (sociodemographic and clinical data) were already specified. We would like to remark that anthropometric measures were never assessed and therefore do not appear in variables assessed. (Please refer to lines 157 to 163 to verify this information).
Variables
Demographic and clinical data collection
Information was obtained through organized interviews using a customized questionnaire designed to collect sociodemographic and clinical data. Clinical variables included the time since diagnosis, tumor stage, family history of BC, surgical procedures, types of treatment received, current medications, presence of metastasis or recurrence, menopausal status, utilization of psychological or physiotherapy services, and lifestyle factors such as tobacco and alcohol use.
- In the results of the abstract, include information about the “p” values in relation to the statistical analyses performed. If possible, also mention which main statistical analyses were performed in the methodology.
Author response:
Thank you for your insightful comment. Regarding the inclusion of p values in the abstract, we acknowledge the importance of providing statistical details. However, given that our study includes three groups, specifying pairwise comparisons for each variable would considerably increase the length of the abstract, making it difficult to stay within the word limit. Instead, we have opted to report a global* p value to summarize the main findings while ensuring clarity and conciseness (Please refer to lines 35 to 41).
*Note: When referring to a “global” p-value, what we want to transmit is that the results paragraph depicts how the lowest self-perceived cardiorespiratory fitness group compares to the other two groups. This global p-value makes reference to all the variable aforementioned in the sentence being statistically significant when comparing the very poor/poor to both the average and good/very good groups.
Additionally, to enhance the statistical robustness of the abstract, we have incorporated the p values and beta coefficients for the regression analysis. For a more detailed breakdown of statistical comparisons, we encourage referring to the results section in the main document, where all specific p values are comprehensively reported (Please refer to lines 35 to 41).
We appreciate your suggestion and hope that this approach maintains the necessary balance between conciseness and statistical rigor. Please let us know if further refinements are needed.
Results: Among the 80 LTBCSs, 35% of LTBCSs reported very poor/poor self-perceived cardiorespiratory fitness, 35% reported average levels, and 30% reported good/very good levels. Individuals with lower self-perceived cardiorespiratory fitness levels showed significant declines in physical fitness, greater physical inactivity, increased CRF, higher pain levels, and poorer HRQoL (p < 0.05). Regression analysis identified “self-perceived muscle strength" (β = 0.40; p < 0.01) and "nausea and vomiting" (β = -0.37; p < 0.01) as significant predictors of higher self-perceived cardiorespiratory fitness (r²adjusted = .472).
“Methods: A cross-sectional study was carried out involving 80 LTBCSs, divided into three groups according to their self-reported cardiorespiratory fitness levels: very poor/poor (1–2), average (3), and good/very good (4–5). Sociodemographic and clinical data were collected, and the study analyzed variables measured at least five years after diagnosis, focusing on various factors including physical fitness, physical activity (PA) levels, cancer-related fatigue (CRF), mood, pain, and health-related quality of life (HRQoL). ANOVA, Mann-Whitney U, and Chi-square tests were performed, along with correlation and multiple regression analysis. Cohen's d was used to calculate effect sizes”.
- In the conclusion of the abstract, you are talking about the results of the study. I suggest reviewing the conclusion focusing on the objectives and that the information about the results be included in the results topic.
Author response:
Thank you for your valuable feedback. We appreciate your suggestion regarding the conclusion of the abstract. As per your recommendation, we have revised this section to better align with the study objectives and avoid reiterating specific results (Please refer to lines 41 to 47).
In the revised conclusion, we have shifted the focus towards the broader implications of our findings, emphasizing the relevance of self-perceived cardiorespiratory fitness as an important indicator of health outcomes in LTBCSs. Additionally, we highlight the need for integrating both subjective perceptions and objective assessments in the development of rehabilitation strategies to enhance long-term health and quality of life in this population. We believe this revision strengthens the clarity and impact of the abstract, ensuring a more precise distinction between the results and the conclusion. Thank you again for your insightful suggestions.
“Conclusions: These findings highlight the importance of self-perceived cardiorespiratory fitness as a relevant indicator of health outcomes in LTBCSs. Given its association with physical fitness, sedentary behavior, CRF, pain, and HRQoL, assessing patients’ perceptions may provide valuable insights for developing tailored rehabilitation strategies. Future interventions should consider both subjective and objective measures to optimize long-term health and quality of life in this population”.
Keywords
- Avoid repeating the same words that were presented in the title, the function of keywords is to expand the search for the study.
Author response:
Thank you for your suggestion regarding the keywords. We understand that their purpose is to expand the search for the study rather than repeating terms from the title. Based on your feedback, we have revised the keywords to include broader and complementary terms that enhance the discoverability of our work (Please see lines 48 to 49).
“Keywords: Exercise oncology; Rehabilitation; Perceived physical fitness; Health-related quality of life; Long-term survivorship”.
Introduction
- In the introduction, you advocate the self-perceived assessment of the health components of women who have had breast cancer for more than 5 years. However, what I missed in the introduction was seeing results from previous clinical studies that made this assessment with people who had cancer or with other special groups, as well as associating or correlating self-perceived health with physical tests to see if they really follow the same path. Therefore, I believe that including this type of study in the introduction could make the introduction more robust.
Author response:
Thank you very much for your thoughtful comments and suggestions, which I find very valuable. Regarding your observation on the need to include studies addressing the relationship between self-perceived health and physical tests in people with cancer, I would like to clarify that this aspect is already implicitly addressed in the introduction.
In the introduction section, it is mentioned that while there are previous studies exploring the relationship between self-perceived fitness and objective measures in populations such as adolescents, adults, or short-term breast cancer survivors (less than 5 years post-diagnosis) [7-10], research on long-term breast cancer survivors (LTBCSs, more than 5 years post-diagnosis) remains limited. Additionally, the introduction emphasizes that self-assessment, particularly in relation to cardiorespiratory fitness, has been of interest in other populations, and correlational research has shown that non-exercise models incorporating variables such as age, gender, body mass index, resting heart rate, and both self-reported physical fitness and physical activity (PA) levels can reliably predict cardiorespiratory fitness [12-15]. This highlights the importance of self-perceived fitness as a clinical tool, which has been examined in other populations but not yet in LTBCSs, thus creating a gap in the literature that this study aims to address.
Therefore, I believe the introduction already presents the justification for this study in a concise and clear manner, emphasizing the existing research gap without the need to extend the section unnecessarily.
However, if you still feel that this response is insufficient, we are open to discussing the possibility of further expanding the introduction. Nevertheless, we humbly believe this may not be necessary, as the introduction already reflects both the clinical utility of self-perception in the context of health and the fact that there are no studies in LTBCS that have examined this topic. For this reason, we have cited similar studies from other populations where similar assessments have been made.
Once again, thank you for your valuable feedback. (Please see lines 51 to 106).
Note: We wanted to add this in the introduction because we believe it reinforced the message: Additionally, studies in various populations, including adolescents, adults, and short-term BC survivors, have linked higher self-perceived cardiorespiratory fitness to improved physical, mental health, as well as with an enhanced HRQoL [7-10]."
Material and methods
- I could not find the clinical trial registration, please insert the registration number and the website link.
Author response:
Thank you for your comment. As this study is a cross-sectional observational study without any intervention, there was no need for clinical trial registration. The study involved a single assessment of the participants at the time of their visit to the center, and no follow-up or treatment interventions were conducted. As far as we know, a clinical trial registration is not applicable in this case and we have never been asked to provide this information in previous publications done by our research team. We hope this clarifies the situation, and we are happy to provide any further information if needed.
- Please include the study flowchart in the text. Follow the CONSORT 2010 model (https://view.officeapps.live.com/op/view.aspx?src=https%3A%2F%2Fwww.equator-network.org%2Fwp-content%2Fuploads%2F2013%2F09%2FCONSORT-2010-Flow-Diagram-MS-Word.doc&wdOrigin=BROWSELINK).
Author response:
Thank you for your thoughtful comments. I would like to clarify that this study is cross-sectional. As indicated in the link provided, the CONSORT 2010 model requires allocation by intervention. However, no intervention was done in this observational study. The participants were divided into three groups based on their self-perceived cardiorespiratory fitness levels, and this categorization is explicitly commented in the methods section, no randomization was done.
However, seeing as the flowchart is also mentioned as a suggestion in Comment 11, we have added the flowchart as Supplementary Material 1. Furthermore, we felt it was more important to prioritize space for the presentation of other relevant tables and figures, which we considered more critical for the understanding of the results, rather than dedicating space to a flowchart. I hope this clarifies the reasoning behind our decision, and I appreciate your understanding. If you deem it necessary to include this flowchart in the manuscript, we will take it into consideration
- In the methodology, I did not find any information about anthropometric and body composition assessments of the participants, nor did I find any assessment of physical tests to compare with self-reported health information. If this was done, please include it, as these factors affect the perception of health and quality of life, as well as physical fitness. Describe how they were carried out, as well as the brands and scales of the equipment.
Author response:
We appreciate the reviewer’s thoughtful comment regarding the inclusion of anthropometric and body composition assessments, as well as objective physical fitness tests. However, we would like to clarify that the primary objective of our study was to examine the association between self-perceived cardiorespiratory fitness and health outcomes in long-term breast cancer survivors (LTBCSs), as well as to identify its key predictors in women at least five years post-diagnosis. Given this specific aim, our study focused exclusively on self-perception measures rather than objective physical assessments such as body composition analysis or direct cardiorespiratory fitness testing.
While incorporating objective physical fitness measures would have been valuable for comparing self-perceived and actual cardiorespiratory fitness, this was beyond the scope of our study, which specifically aimed to assess subjective perceptions and their relationship with various health-related variables. We acknowledge that including objective measurements could provide additional insights, particularly regarding the accuracy of self-perception in this population.
To address this valid point, we will include an additional limitation in our discussion, highlighting that height, weight, body composition, and objective cardiorespiratory fitness assessments were not included. We will also emphasize that future research could benefit from integrating both subjective and objective measures to further explore their relationship while maintaining the primary focus on self-perception (Please see lines 497 to 511).
Thank you again for this insightful suggestion.
“This study has several limitations. The cut-off points used to classify participants align with previous categorizations [14,19], yet alternative values might have altered the results. Additionally, its cross-sectional design prevents establishing causal relationships between self-perceived cardiorespiratory fitness and physical, mental, and emotional variables, highlighting the need for longitudinal studies. The lack of prior research in LTBCSs using this analytical approach with the IFIS and these specific cut-off points further complicates comparisons. Moreover, anthropometric measurements (e.g., height, weight, and body composition) and objective cardiorespiratory fitness assessments were not included, as the study focused exclusively on self-perception. While this aligns with our primary aim, future research should consider incorporating these objective measures to better understand the relationship between self-perceived and actual fitness levels. Future studies should build on these findings to explore self-perceived cardiorespiratory fitness in relation to health status and other fitness components, such as muscular strength, speed/agility, and flexibility”.
- I am unsure about how the recruitment of people was carried out so that you could distribute them into the three groups. Was it convenient? That is why it is important to present the research flowchart.
Author response:
Thank you for your comment. We would like to clarify that our study follows a cross-sectional design, and as such, participants were not assigned to groups through a randomization process. Recruitment and group distribution is detailed in the "Design and Participants" section of the Methods (Recruitment: Lines 113 to 116; Group classification: Lines 144 to 147).
Regarding group classification, participants were not pre-assigned to specific groups during recruitment. After reaching the planned sample size (n = 80), we categorized them retrospectively based on their self-perceived cardiorespiratory fitness, assessed using the International Fitness Scale (IFIS). Once the sample size was reached, and patient outcomes showed 3 homogenous groups, assessments were stopped and analysis commenced. Since no participants were excluded after initial recruitment and the study focused on self-perceived fitness rather than an interventional design, we initially considered that a flowchart was not essential.
However, in response to your suggestion and to enhance methodological transparency, we have now included a flowchart as Supplementary Material 1, detailing the recruitment and classification process. This flowchart is now referenced in the Methods section (Please see lines Lines 113 to 116).
We appreciate your valuable feedback and hope this addition clarifies any concerns.
“A total of 80 LTBCSs participated in this study. Eligible participants were identified and recruited through the oncology department of the University Hospital Complex of Granada during the specified study period. Additional details can be found in Supplementary Figure 1”.
Results
- Figure 1 is not in very good resolution, please improve the image quality and font size.
Author response:
Thank you for your observation. We believe that the issue with Figure 1's resolution may have resulted from the journal's platform compressing the uploaded file. To address this, we have re-uploaded the figure in its original high-quality format within the revised submission.
We hope that it now displays correctly, but if the issue persists, we will wait for the journal’s feedback to determine how to proceed in case the editorial system automatically compresses the file again.
We appreciate your attention to detail and your valuable feedback.
- After figure 1, an untitled frame appears, please adjust it. I believe it could be table 4, however, it was created in frame format.
Author response:
Thank you for bringing this to our attention. We have reviewed the issue, and it was likely due to the original format of Table 4 before modifications. Since Reviewer 1 also provided feedback on this matter, we have made the necessary adjustments, and we believe that the formatting should now display correctly.
However, if the issue persists or if we have not fully addressed your concern, we remain open to further feedback and will make any additional changes as needed.
Discussion
- The discussion should be strengthened with more clinical studies that evaluated these same variables and included studies with people who performed physical tests to prove whether perception can also be portrayed in people's assessments.
Author response:
Thank you very much for your comment. With regards to studies with people who performed physical tests to prove whether perception can also be portrayed in people's assessments, we thought it necessary to add additional information in the discussion with reference to studies that have explored that self-perceived results are similar to those obtained by objective assessments (Lines X to X). However, this information provided is not comparable to long-term breast cancer survivors seeing as these variables have, to date, not been studied in this population.
“As for applicability to objective measures, two studies have shown that higher self-reported fitness levels were associated with a higher objectively measured fitness level [37,38]. However, the lack of specificity regarding LTBCSs in these three previous studies limits direct comparison.”
Reference:
- Obling KH, Hansen A-LS, Overgaard K, et al. (2015) Association between self-reported and objectively measured physical fitness level in a middle-aged population in primary care. Prev Med Rep 2: 462-466. https://doi.org/10.1016/j.pmedr.2015.05.010.
- Aanstad A (2022) Self-perceived and self-tested endurance: Association with objective measure. Percept Mot Skills 129:1492-1503. https://doi.org/10.1177/00315125221107852.
As for strengthening the discussion with more clinical studies that evaluate the same variables, we believe that we have made the most of the current literature that is available regarding the variables that we have measured seeing as all individual paragraphs of the discussion, each one specific to a variable, is compared at some point with available evidence. Specifically:
- “Similar trends have been observed in a study involving a mixed population of short- and long-term cancer survivors across various cancer type, where lower self-perceived fitness correlated with reduced functional performance and capacity [36]. However, the lack of specificity regarding LTBCSs in that study limits direct comparison (Lines 412 to 418).”
- “This aligns with prior research showing that individuals perceiving themselves as fitter engage in more PA [39] (Lines 420 to 421).”
- “This aligns with research linking lower self-perceived fitness to higher CRF levels and poorer well-being in cancer survivors [42] (Lines 430 to 431).”
- “While previous studies have linked objective cardiorespiratory fitness measures to CRF outcomes in BC survivors [43], evidence specifically addressing self-perceived cardiorespiratory fitness in LTBCSs remains limited (Lines 434 to 436).”
- “While previous studies in adolescents and short-term BC survivors have reported associations between higher self-perceived fitness and improved mood or lower levels of psychological distress [10,46-49], evidence within long-term BC survivorship remains scarce (Lines 434 to 438).”
- “This aligns with research showing that a home-based walking intervention improved cardiorespiratory fitness and reduced pain in patients undergoing treatment for solid tumors [50]. Although that study did not focus on self-perceived fitness, it suggests that PA may help alleviate pain. Similarly, studies in healthy adults link better self-perceived fitness to lower pain sensitivity [51] (Lines 448 to 451).”
- “While research on self-perceived cardiorespiratory fitness and HRQoL in LTBCSs is limited, studies in other populations suggest a strong link between perceived fitness and well-being. For instance, better self-perceived fitness in adolescents correlates with improved psychological well-being and lower distress [52]. In healthy adults, interventions targeting self-perceived fitness have been linked to physical and emotional health benefits [53]. Among BC survivors, higher self-perceived physical fitness has been associated with better HRQoL and emotional functioning [10]. However, no studies have specifically examined self-perceived cardiorespiratory fitness and HRQoL in LTBCSs, highlighting the need for further research (Lines 458 to 466).”
- “These findings align with previous research indicating that better perceptions of physical fitness are linked to enhanced HRQoL in BC survivors [54] (Lines 471 to 472).”
- “Similar findings have been reported regarding self-perceived physical fitness and its impact on mental health in cancer survivors [36]. However, said study [36] did not account for cancer type, treatment, or differences between short- and long-term survivors. Since symptom severity and patient needs vary by survivorship stage [55,56], further research is needed to assess the role of self-perceived cardiorespiratory fitness in LTBCSs (Lines 484 to 489).”
Additionally, the introduction also reflects comparison to available clinical studies:
- “Among the various components of physical fitness, cardiorespiratory fitness stands out as a key indicator of overall health and health-related quality of life (HRQoL) [6]. Additionally, studies in various populations, including adolescents, adults, and short-term BC survivors, have linked higher self-perceived cardiorespiratory fitness to improved physical, mental health, as well as with an enhanced HRQoL [7-10]. It is also associated with a greater likelihood of engaging in PA and adopting healthier lifestyles [8].”
- “As a practical alternative, self-reported measures have been developed to estimate cardiorespiratory fitness without requiring exhaustive testing. Research has shown that non-exercise models incorporating variables such as age, gender, body mass index, resting heart rate, and both self-reported physical fitness and PA levels can reliably predict cardiorespiratory fitness [12-15].”
- Regarding the limitations of the study, I believe that a major limitation is the fact that you did not perform physical tests on people. This prevents us from knowing whether people were reporting what their bodies could respond to when challenged in a physical test. This question raises the question: if these people were previously subjected to physical tests to assess strength, muscular endurance, flexibility and cardiorespiratory capacity, would they have answered the questionnaire in the same way?
Author response:
Thank you very much for your suggestion. We want to reiterate, as mentioned in comments 7 and 10, that although we find the question very interesting, our study focused solely on self-perceived outcomes using a validated tool. Nevertheless, we have made additional remarks at the end of the limitations paragraph, incorporating that it would be interesting to see the comparison of our results with objective results (Lines 497 to 511).
“This study has several limitations. The cut-off points used to classify participants align with previous categorizations [14,19], yet alternative values might have altered the results. Additionally, its cross-sectional design prevents establishing causal relation-ships between self-perceived cardiorespiratory fitness and physical, mental, and emotional variables, highlighting the need for longitudinal studies. The lack of prior re-search in LTBCSs using this analytical approach with the IFIS and these specific cut-off points further complicates comparisons. Future studies should build on these findings to explore self-perceived cardiorespiratory fitness in relation to health status and other fitness components, such as muscular strength, speed/agility, and flexibility. Moreover, anthropometric measurements (e.g., height, weight, and body composition) and objective cardiorespiratory fitness assessments were not included, as the study focused exclusively on self-perception. While this aligns with our primary aim, future research should consider incorporating these objective measures to better understand the relationship between self-perceived and actual fitness levels. This questions whether self-perceived results correspond to objective results or if participants under- or over-estimate their capabilities.”
Conclusions
- Please try to make the conclusion more direct in relation to the objectives.
Author response:
Thank you very much for your appreciation. Reviewer 1 also raised similar concerns regarding the conclusion, and after carefully considering the feedback from both reviewers, we have revised the conclusion to ensure it more directly addresses the study’s objectives. We believe that the updated version now better reflects the overall content of the manuscript and provides a clear response to the research questions posed. The revised conclusion can be found on lines 523 to 539.
“In conclusion, only 30% of LTBCSs reported having good/very good levels of self-perceived cardiorespiratory fitness. Furthermore, LTBCSs with lower self-perceived cardiorespiratory fitness experienced a marked deterioration in fitness levels and increased sedentary behavior, as well as increased CRF, higher pain levels, and reduced HRQoL five or more years post-diagnosis. The combination of "self-perceived muscle strength" and "nausea and vomiting" explained 47.2% of the variance in self-perceived cardiorespiratory fitness among LTBCSs.
These findings underscore the critical role of patients’ perceptions in the development of tailored rehabilitation strategies, emphasizing the necessity of integrating both subjective experiences and objective assessments to optimize long-term health outcomes and HRQoL in LTBCSs. From a practical standpoint, these results highlight the need for healthcare professionals to incorporate regular assessments of self-perceived fitness into survivorship care plans. Targeted interventions, such as personalized exercise programs and symptom management strategies, could help mitigate the negative effects of low self-perceived fitness and enhance overall well-being. Future research should explore the effectiveness of structured physical activity interventions in improving both perceived and objective fitness levels in this population”.
References
- It has been found that there are many references that are more than 5 years old, please try to replace them with more recent references.
Author response:
Thank you for your observation. We have updated the following references:
New reference 2:
- Newell, A., Malhotra, J., Raoof, E. et al (2024) Catalyzing Progress: a Comprehensive Review of Cancer Rehabilitation Education for Rehabilitation Specialists. Curr Phys Med Rehabil Rep 12, 177–185. https://doi.org/10.1007/s40141-024-00441-x
New reference 3:
- Kochman M, Kielar A, Kasprzak M, Maruszczak K, Kasperek W (2023) The Relationship between Self-Rated Health and Physical Fitness in Polish Youth. Healthcare (Basel) 21;12(1):24. doi: 10.3390/healthcare12010024.
New reference 5:
- Aandstad, A. (2022). Self-Perceived and Self-Tested Endurance: Associations with Objective Measures. Perceptual and Motor Skills 129(5), 1492-1503. https://doi.org/10.1177/00315125221107852
New reference 11:
- Campbell KL, Winters-Stone KM, Wiskemann J, May AM, et al (2019) Exercise Guidelines for Cancer Survivors: Consensus Statement from International Multidisciplinary Roundtable. Med Sci Sports Exerc 51(11):2375-2390. doi: 10.1249/MSS.0000000000002116.
New reference 31:
- Cerezo O, Oñate-Ocaña LF, Arrieta-Joffe P, et al (2012) Validation of the Mexican-Spanish version of the EORTC QLQ-C30 and BR23 questionnaires to assess health-related quality of life in Mexican women with breast cancer. Eur J Cancer Care (Engl) 21(5):684-91. doi: 10.1111/j.1365-2354.2012.01336.x.
The remaining references weren’t changed due to either lack of more recent information on the subject (self-perceived and/or in long-term breast cancer survivors), because they are pertinent to the validity and reliability of tools used, or were commonly used sources for definitions and terms. We want to note that even though reference 31 was published more than 5 years ago, it is an updated version of the previous one and there is no more recent option available so far.
The author responsible for correspondence is:
Maria Figueroa Mayordomo
Department of Physiotherapy, Faculty of Health Sciences,
European University of Valencia,
46112 Valencia, Spain
Tel +34-663746655
Sincerely,
Maria Figueroa Mayordomo

Round 2
Reviewer 2 Report
Comments and Suggestions for Authors
Dear editor and authors, After evaluating the adjustments made to the manuscript, my suggestion is that the work be accepted.